# MgO–ZrO_2_ Ceramic Composites for Silicomanganese Production

**DOI:** 10.3390/ma15072421

**Published:** 2022-03-25

**Authors:** Cristian Gómez-Rodríguez, Linda Viviana García-Quiñonez, Josué Amilcar Aguilar-Martínez, Guadalupe Alan Castillo-Rodríguez, Edén Amaral Rodríguez-Castellanos, Jesús Fernando López-Perales, María Isabel Mendivil-Palma, Luis Felipe Verdeja, Daniel Fernández-González

**Affiliations:** 1Faculty of Engineering, University of Veracruz, Coatzacoalcos 96535, Mexico; crisgomez@uv.mx; 2Departamento de Ciencia de los Materiales e Ingeniería Metalúrgica, Escuela de Minas, Energía y Materiales, Universidad de Oviedo, 33004 Oviedo, Asturias, Spain; lfv@uniovi.es; 3CONACYT-Centro de Investigación Científica y de Educación Superior de Ensenada B.C. (CICESE), Unidad Monterrey, Apodaca 66629, Mexico; 4Facultad de Ingeniería Mecánica y Eléctrica (FIME), Universidad Autónoma de Nuevo León (UANL), San Nicolás de los Garza 66450, Mexico; josue.aguilarmrt@uanl.edu.mx (J.A.A.-M.); alan.castillo.rdz@gmail.com (G.A.C.-R.); eden.rodriguezcs@uanl.edu.mx (E.A.R.-C.); jlopezp@uanl.edu.mx (J.F.L.-P.); 5Centro de Investigación en Materiales Avanzados, S.C. (CIMAV-Sede Monterrey), Alianza Norte 202, Parque de Investigación e Innovación Tecnológica, Apodaca 66600, Mexico; maria.mendivil@cimav.edu.mx; 6Centro de Investigación en Nanomateriales y Nanotecnología (CINN), Consejo Superior de Investigaciones Científicas (CSIC), Universidad de Oviedo (UO), Principado de Asturias (PA), Avda. de la Vega, 4-6, 33940 San Martín del Rey Aurelio, Asturias, Spain

**Keywords:** sintering, slag, corrosion, spectroscopy, chemical properties, mechanical properties, MgO, silicomanganese, nanomaterials, refractories, ceramics

## Abstract

The deterioration of the refractory lining represents a significant problem for the smooth operation in the ferroalloys industry, particularly in the production of silicomanganese, due to the periodic requirements of substitution of the damaged refractory. Within this context, magnesia refractories are commonly employed in the critical zones of the furnaces used in silicomanganese production since the slag involved in the process has a basic character. The behavior of MgO–ZrO_2_ ceramic composites with different ZrO_2_ nanoparticles (0, 1, 3, and 5 wt.%) contents in the presence of silicomanganese slags is proposed in this manuscript. XPS, XRD and SEM–EDX were used to evaluate the properties of the ceramic composite against the silicomanganese slag. The static corrosion test was used to evaluate the corrosion of the refractory. Results suggest that corrosion is controlled by the change in slag viscosity due to the reaction between CaZrO_3_ and the melted slag. Besides, ZrO_2_ nanoparticles located at both triple points and grain boundaries act as a barrier for the slag advance within the refractory. The utilization of MgO refractories with ZrO_2_ nanoparticles can extend the life of furnaces used to produce silicomanganese.

## 1. Introduction

Materials used in furnaces must withstand the adverse conditions during the operation at high temperatures required to produce metals, which include: (i) corrosion and erosion by solids, liquids, fumes, and gases; (ii) thermo-mechanical loads generated in different parts of the furnace [1,2,3,4,5]. Refractory ceramics are used with this aim in the refractory lining of furnaces, and, within this context, around 70% of the worldwide refractory production is used by the metallurgy industry [2,6]. Basic refractory ceramics (mainly magnesia ceramics) are, from those used in metallurgy, one of the most important families of refractories, since they are used in metallurgical ladles, electric arc furnaces (EAF), basic oxygen furnace (BOF), and in special furnaces for non-ferrous metallurgy. The problem arises from the soon deterioration (wear, cracks, spalling due to chemical corrosion produced during the metal smelting) of the refractory resulting from operation under extreme conditions, which requires year by year, from large amounts of money for maintenance or total replacement of damaged refractory linings.

Silicomanganese (17–20% Si and 1.5–2.0% C) is mainly consumed in electric arc furnace steel production [7,8,9,10]. Its utilization is expected to advance at a faster rate than ferromanganese and ferrosilicon because it adds less phosphorus, carbon, aluminum, and nitrogen [7,8,9,10] to the steel. It is also an effective deoxidizer, which results in cleaner steel. Low and ultra-low carbon silicomanganese alloys (26–31% Si and 0.05–0.5% C) are used in the manufacture of stainless steels and special steels [7,8,9,10].

Silicomanganese alloy is produced by carbothermal reduction in raw materials (manganese ores, quartzite, (Fe)Si-remelts, coke (1.5–2%), and high carbon ferromanganese production slag (35–45% MnO), since both types of alloys are produced in the same factory) in a Submerged Arc Furnace (SAF) at temperatures within 1600 °C and 1700 °C [11]. One of the critical issues during the silicomanganese production is the refractory lining damage in the tap-hole area and the furnace hearth due to the wear and chemical corrosion caused by molten metal and slag [11], which involves a non-smooth production process and economic and environmental impacts resulting from the periodic replacement of the refractory. Silicomanganese slag (around 1225 kg slag per ton) is disregarded when it contains 6 to 12% of dissolved MnO, where the other main constituents are SiO_2_ (38–44%), CaO (20–35%), MgO (5–15%), and Al_2_O_3_ (10–25%). Within this context, little literature is available about the deterioration of the refractories in the presence of silicomanganese slag, and this is limited to SiC-based and C-based refractories [11,12,13,14]. However, the improvement of refractories for the SiMn alloy production is essential due to the technological importance (23.5 Mt are expected for 2025) of this material alloy in the steelmaking process [7,8,9,10].

Historically, MgO-based refractories have fulfilled the requirements of the metallurgical and steel industries. The addition of different additives, particularly oxides, has been subject of research for more than 10 years. C [15,16,17], Al_2_O_3_ [18], Cr_2_O_3_ [19,20,21], MgAl_2_O_4_ [22,23], ZrSiO_4_ [24], ZrO_2_ [25], TiO_2_ [26], FeAl_2_O_4_ [27], or CaZrO_3_ [28] were added to improve the properties of MgO-based refractories, and considering nanoparticles, it is possible to point out Fe_2_O_3_ [29,30], Al_2_O_3_ [30,31,32], ZrO_2_ [33,34], ZrSiO_4_ [35,36], C [37,38], MgAl_2_O_4_ [39], TiO_2_ [40,41], Cr_2_O_3_ [42,43], and SiO_2_ [44].

This manuscript presents an innovative study about the behavior of MgO reinforced with ZrO_2_ nanoparticles in the presence of silicomanganese slag. The influence of different contents of ZrO_2_ nanoparticles (0, 1, 3, and 5 wt.%) is discussed together with the effect of the method of obtaining the green compacts (cold uniaxial pressing or cold uniaxial pressing and cold isostatic pressing) and also the sintering temperature (1550 °C or 1650 °C). The results of this research work could represent a potential benefit for the industry of SiMn alloy production from the environmental and economic points of view.

## 2. Materials and Methods

### 2.1. Materials

Sintered magnesia (MgO) with an average particle size < 45 µm (98.5% MgO, 1% CaO, 0.2% SiO_2_, 0.15% Al_2_O_3_, 0.13% Fe_2_O_3_, 0.01% B_2_O_3_, 3.48–3.52 g/cm^3^ bulk density, supplied by Grupo Peñoles company (Laguna del Rey, Coahuila, México) was used as starting material. It is well known that the fine fraction is considered the weakest constituent of a refractory matrix. Therefore, it has to be reinforced by a strong bond development. The bonding strength represents one of the main microstructural characteristics contributing to the reliable refractory matrix establishment. Increasing the bonding strength, the resistance against many kinds of stresses during the performance, and structural spalling would be improved. Fine fraction study is important since it is the one that has the highest reactivity in a refractory system. Besides, the aggregates are the main refractory constituent that supports the mechanical, thermal, and chemical changes in a system. On the other hand, there are many studies focused on the fine elements of a refractory (matrix and bonding structure), as was cited below. Through these studies, it was found that reducing the particle size (<45 µm) helps the thermal sintering process by improving the morphology and microstructure of sintered composites with a beneficial impact on the mechanical, physical, and chemical properties [17,28,30,34].

High purity zirconia (ZrO_2_) nanoparticles (<100 nm, >99.9% purity, 5.89 g/cm^3^ relative density, ≥25 m^2^/g specific surface area, supplied by Skyspring Nanomaterials Inc., (Houston, TX, USA) were also used as starting raw material. ZrO_2_ nanoparticles used in this manuscript had a monoclinic structure. Zirconium oxide (ZrO_2_) is a phase that characterizes by exhibiting three polymorphic transformations. It is monoclinic from room temperature up to 1170 °C, tetragonal from 1171 °C to 2370 °C, and cubic above 2370 °C and until the melting point (2715 °C). It is well known that there are several phases that, as the MgO, CaO, and Y_2_O_3_ are adequate to stabilize a single crystalline structure of ZrO_2_, that is to say, they would remain with a certain crystalline structure form from room temperature to the melting point. Thus, without polymorphic changes (monoclinic–tetragonal–cubic, or vice versa) and, therefore, volume changes that could involve cracks in the parts.

Silicomanganese slag from the metallurgical process was used to assess the chemical resistance of experimental refractory compositions (wt.%): 26.88% SiO_2_, 24.85% CaO, 24.56% MnO, 12.41% Al_2_O_3_, 4.55% MgO, 1.21% BaO, 0.76% Na_2_O, 0.80 K_2_O, and balanced others (Fe, P, Ti, Sr). Silicomanganese slag was milled and screened to <38 µm. The slag chemical composition was determined by an Axios, a PANalytical wavelength-dispersive X-ray fluorescence spectrometer (Servicios Científico-Técnicos, Universidad de Oviedo, Oviedo, Asturias, Spain) with an Rh-anode X-ray tube as a radiation source (4 kW as maximum power). Samples were measured in a vacuum with a 15–50 eV energy resolution. For the quantitative analysis of the spectra, the Omnian software was used.

The slag is considered basic when BI > 1 considering the basicity index BI = (% CaO + % MgO)/(% SiO_2_) [45,46]. Therefore, the SiMn slag is a basic slag (1.09), so the chemical compatibility with MgO-based refractories is high.

### 2.2. Sample Preparation

Weight percentages of ZrO_2_ nanoparticles were added to magnesia powders considering the following relation: (100 − X) wt.% MgO + X wt.% of ZrO_2_, where X = 0, 1, 3 and 5. Table 1 shows the experimental design and physical properties carried out in this research. Bulk density and porosity tests were developed in accordance with the ASTM-C830-00 Standard and as described in the same standard [45].

The nanoparticle’s dispersion was obtained using Zephrim PD3315 as a dispersant and acetone as a dispersion medium. A 1/10 ratio (nanoparticles/dispersant) was used. Afterward, the powders with the specific nanoparticle concentration and ethylene glycol (2 wt.% of the total mass) as a binder were mixed for 15 min by a mechanical method using an Alghamix II-Zhermack mechanical mixer (Facultad de Ingeniería Mecánica y Eléctrica, San Nicolás de los Garza, Nuevo León, México) at 100 rpm to obtain a suitable homogeneity.

Then, the resulting powder mixtures were formed in green cylinder samples (25 mm in diameter and 30 mm in height) using two options: cold uniaxial pressing (CUP) at 100 MPa for 2 min; cold uniaxial pressing (CUP) at 100 MPa for 2 min and cold isostatic pressing (CIP) in an autoclave (Autoclave Engineers, Inc. P-419 (located in Centro de Investigacion en Nanoteriales y Nanotecnologia (CINN), L’Entregu, Asturias, Spain) at 200 MPa for 5 min.

All the experimental samples were dried in a muffle (Departamento de Ciencia de los Materiales e Ingeniería Metalúrgica, Escuela de Minas, Energía y Materiales, Universidad de Oviedo, Oviedo, Asturias, Spain) at 250 °C for 24 h to evaporate moisture. After the drying process, sintering was carried out in a Lindberg Blue M/1700 Thermo Fisher Scientific electric furnace (Escuela Politécnica de Mieres, Universidad de Oviedo, Mieres, Asturias, Spain) using a heating rate of 5 °C/min with a dwell time of 4 h at maximum temperature (1550 °C or 1650 °C, depending on the sample). Cooling down to room temperature was carried out in the furnace.

### 2.3. Methods

#### 2.3.1. Microstructural Analysis

The microstructural analysis was carried out using a Philips X’Pert diffractometer (Servicios Científico-Técnicos, Universidad de Oviedo, Oviedo, Asturias, Spain) equipped with Cu anode tube and Kα1 radiation (λ = 1.54056 Å) to investigate the crystallographic information, in the range of 2Ɵ from 15° to 90° at a scan speed of 1°/s. Data analysis and the peak profile fitting were carried out using the XPowder program. The main crystalline phases of the silicomanganese slag are gehlenite (Ca_2_Al(SiAl)O_7_), kirschsteinite (CaFe^2+^SiO_4_), and fukalite (Ca_4_[Si_2_O_6_][CO_3_](OH)_2_) with an average phase percentage of around 22%. Melilite phase (Ca, Na)_2_(Al, Mg) (Si, Al)_2_O_7_) represents a phase percentage of 14%, and other phases that represent less than 10% were detected (rhodochrosite—MnCO_3_, goethite—(α-Fe^3+^O(OH), tephroite—Mn_2_SiO_4_ and mozartite—CaMn^3+^(SiO_4_)(OH)).

#### 2.3.2. Morphological Analysis

The morphological analysis of the MgO refractory reinforced with ZrO_2_ nanoparticles, sintered at 1550 °C and 1650 °C, was carried out by a JEOL-6610LV scanning electron microscopy (Servicios Científico-Técnicos, Universidad de Oviedo, Oviedo, Asturias, Spain) equipped with an electron disperse X-ray spectroscopy (EDX) detector (Inca energy-200).

#### 2.3.3. Mechanical Properties

Mechanical resistance of the composites was determined by cold crushing strength in specimens of sintered samples corresponding to cylindrical specimens with a diameter of 24 mm and a height of 11 mm.

An ELE-INTERNATIONAL hydraulic universal press machine (Departamento de Ciencia de los Materiales e Ingeniería Metalúrgica, Escuela de Minas, Energía y Materiales, Universidad de Oviedo, Oviedo, Asturias, Spain), model ABR-AUTO with a maximum load set at 200 tons, was used for the evaluation. The compressive load was applied parallel to the cylindrical samples at a speed of 50 kgf/s and at room temperature.

#### 2.3.4. Chemical State Analyses

The elemental composition and oxidation state analyses of sintered samples were carried out by a Thermo Scientific X-ray photoelectron spectroscopy (XPS, Facultad de Ingeniería Mecánica y Eléctrica, San Nicolás de los Garza, Nuevo León, México) using a K-alpha X-ray photoelectron spectrometer system. The samples were excited by a monochromatized Al Kα X-ray radiation of energy of 1486.6 eV.

Cleaning by a soft surface etching step was performed to remove superficial impurities from the sample during the analysis. Binding energies of all the peaks were corrected using C 1 s energy at 284.6 eV corresponding to adventitious carbon. Moreover, the charge compensation was corrected by the flood gun associated with the spectrometer. The peaks were deconvoluted using a Shirley type background calculation and peak fitting using the Gaussian–Lorentzian sum function.

#### 2.3.5. Chemical Properties

A measurement of slag penetration by a static corrosion test (finger test method) was performed to determine the chemical resistance of the sintered refractory samples against SiMn slag corrosion. The test consists of drilling a hole (~3.5 mm in diameter with 2.5 mm in depth) in the center of the upper face of the cylindrical sample and filling it with powder slag (5 g). The sintered samples analyzed in this chemical test were 25 mm in diameter and 6 mm in height. The experiment was carried out in an electric furnace for 4 h at 1550 °C. Both the heating and cooling rates were 5 °C/min. Afterward, samples were transversely cut using a diamond disc. After cutting, the surfaces of interest were polished for microscopy evaluation.

Figure 1 shows a scheme of the slag penetration analysis carried out in the refractories. Points A, B, and C represent zones where the morphological analyses by SEM were carried out. These zones correspond to the slag zone (top area), interphase slag/refractory (middle area), and the refractory zone (bottom area). The dotted red line indicates the path followed in all the samples corresponding to the chemical semi-quantitative analysis of elements (wt.% ) by the EDX technique.

## 3. Results and Discussion

### 3.1. Microstructural Analysis

Figure 2 shows the XRD patterns of the reference sample (100 wt.% of MgO) and the MgO sample containing 5 wt.% of ZrO_2_ nanoparticles. This XRD analyses compare the influence of the sintering temperature (1550 °C and 1650 °C) as well as the effect of the method of obtaining the green compacts (CUP or CUP + CIP). These two compositions were chosen as the most representative to identify the crystalline phases in the samples.

As can be observed, the reference samples (100 wt.% of MgO) exhibited in all conditions the same crystalline phase (2nd and 4th diagrams), corresponding to periclase (ICDD 45-0946). The patterns show the characteristic peaks located in 2Ɵ equal to 36.93°, 42.91°, 62.3°, 74.64°, and 78.63° with preferential orientations in the crystallographic planes in (111), (200), (220), (311), and (222), respectively.

Meanwhile, samples with 5 wt.% of ZrO_2_ nanoparticles (1st and 3rd diagrams) display diffraction planes corresponding to magnesium oxide (MgO) (ICDD 45-0946). Moreover, the zirconium oxide phase (ZrO_2_) (ICDD 50-1089) is identified in 2Ɵ equal to 30.27°, 35.25°, 50.37°, and 60.20° with planes diffracted in (011), (110), (112), and (121). Moreover, it was observed in this research that the ZrO_2_ nanoparticles were completely surrounded by MgO particles (matrix constituent). In agreement with the MgO–ZrO_2_ binary diagram, the ZrO_2_ tetragonal phase admits 10% of MgO, and the cubic phase admits up to 27% MgO in solid solution. Thus, there was a change from monoclinic to tetragonal when the composites were sintered at 1600 °C, and this crystallographic change was maintained down to room temperature as the MgO acted as a stabilizer. This can be checked in the X-ray diffraction results in Figure 2 as with 5 wt.% of ZrO_2_, peaks corresponding to the tetragonal ZrO_2_ were observed (ICDD 50-1089). Small peaks with low-intensity corresponding to the calcium zirconate phase (CaZrO_3_) (ICDD 35-0790) were detected. This phase is attributed to the in situ reaction between CaO (phase detected as an impurity of MgO) and the additions of ZrO_2_ nanoparticles.

### 3.2. Morphological Analyses

Figure 3 shows scanning electron microscope images of the reference sample (100 wt.% of MgO) and of the sample with 5 wt.% of ZrO_2_ nanoparticles formed by CUP or CUP + CIP and sintered at 1550 °C or 1650 °C.

Figure 3a,b shows the microstructures of MgO samples formed by CUP or CUP + CIP (XL9 and XL13 samples, sintered at 1650 °C, respectively), where MgO grains (dark gray phase) and small CaO particles (blurred white color phase with a particle size ~3 µm) are observed. The MgO aggregates (10 to 40 µm in size) have an angular morphology, while the MgO particles that form the matrix have a quasi-spherical morphology (<5 µm in size). The CaO particles also have an angular morphology.

Figure 3c,d shows the microstructures of the sample with 5 wt.% of ZrO_2_ nanoparticles formed by CUP or CUP + CIP techniques and sintered at 1650 °C (XL12, and XL16 samples, respectively). Three main phases corresponding to magnesia (MgO), zirconia (ZrO_2_) (circular morphology, bright, intense white color with a particle size < 3 µm), and calcium zirconate (CaZrO_3)_ (rounded light gray particles with a particles size around 5 µm), were observed. As can be observed, ZrO_2_ nanoparticles and CaZrO_3_ particles are at the grain boundaries and triple points. This microstructural characteristic might lead to the development of a pinning effect that allows not only to reduce the porosity but also to increase the density. The XL16 sample (Figure 3d) developed the highest density value (3.05 g/cm^3^) and the lowest porosity percentage (14.48%).

Figure 3e,f, corresponds to the microstructure of MgO samples sintered at 1550 °C (formed by CUP or CUP + CIP techniques, respectively), where MgO grains (dark gray phase) have an irregular shape. Moreover, some CaO particles (soft gray phase) are observed as impurities from raw material. In both microstructures (XL1 and XL5 samples), it is evident a deficit of cohesion between particles develops a dense matrix. This feature opens the opportunity to use a bond phase that allows a better cohesion between particles.

Similar microstructural characteristics were observed in 100 wt.% MgO samples sintered at 1550 and 1650 °C. The highest density (2.71 g/cm^3^) and the lowest porosity (26.24%) were reached by the XL13 sample (100 wt.% MgO) formed by CUP + CIP and sintered at 1650 °C.

Figure 3g,h shows the microstructure of the MgO sample with 5 wt.% ZrO_2_ nanoparticles formed by CUP or CUP + CIP techniques and sintered at 1550 °C (XL4 and XL8 samples, respectively). The microstructural analysis revealed the same phases as those identified in the samples sintered at 1650 °C, i.e., MgO, ZrO_2_, and CaZrO_3_. However, the presence of CaZrO_3_ is lower compared with the samples sintered at 1650 °C. The ZrO_2_ nanoparticles and CaZrO_3_ particles are also located at the grain boundaries and triple points. Figure 3i shows the shape of the phases detected in the XL4 sample: Point 1: CaO, it is possible to see that they are bigger than the nanoparticles, and they appear in blurred white color. Point 2: CaZrO_3_, this phase appears with acicular shape with a color intermediate between the bright, intense white points of ZrO_2_ and the blurred white color of the CaO. Point 3: ZrO_2_ nanoparticles, circular morphology, bright, intense white color.

The difference in density and porosity between the MgO sample that contains 5 wt.% of ZrO_2_ nanoparticles sintered at 1550 and 1650 °C (formed by CUP or CUP + CIP techniques) is remarkable. The combined effect of temperature and forming method leads to higher densification and higher reduction in porosity. Besides, a pinning effect through the addition of ZrO_2_ nanoparticles might contribute to reaching a higher density and an effective porosity reduction.

In general, some mechanisms might promote the densification of the MgO microstructure through the addition of ZrO_2_ nanoparticles [47]: (i) an in situ CaZrO_3_ phase formation with a higher density (4.95 g/cm^3^) than that of the matrix of MgO (3.58 g/cm^3^), (ii) the development of a ceramic bonding through the CaZrO_3_ formation, (iii) a pinning effect due to the specific location of ZrO_2_ and CaZrO_3_ at the grain boundaries and triple points. It is well known that the pinning effect contributes to the porosity elimination, and thus higher densification is obtained. The pinning effect (through ZrO_2_ and CaZrO_3_) avoids the grain boundary movement, allowing the porosity elimination. However, that phenomenon does not mean that the grain size cannot grow since all that is happening during sintering is cohesion between particles that leads to the grain growth. That is why MgO grains grow through the pinning effect. Below we demonstrated this effect graphically. Details of the mechanism are indicated in Figure 4.

### 3.3. Mechanical Properties

Figure 5 shows the variation in the cold crushing strength of the magnesia samples reinforced with ZrO_2_ nanoparticles (1, 3, 5 wt.%) and the reference sample of MgO (100 wt.% of MgO). The increase in the ZrO_2_ nanoparticles content leads to an increase in mechanical resistance since, in all compositions, an increasing tendency in CCS (Cold Crushing Strength) values can be observed. The sample was made with 100 wt.% MgO formed by CUP and sintered at 1550 °C (XL1, reference sample) reached a mechanical resistance of 60.94 MPa. The sample with 5 wt.% of ZrO_2_ nanoparticles formed by CUP and sintered at 1550 °C reached the maximum value of CCS (180.32 MPa, corresponding to the XL4 sample), with an improvement of 66.2% with respect to the reference sample of 100 wt.% MgO, curve 1. Meanwhile, the samples formed by CUP + CIP and sintered at 1550 °C (curve 2) were more sensitive to the forming method since 5 wt.% of ZrO_2_ nanoparticles in the composite resulted in a remarkable improvement of 80.44% with respect to the reference sample of 100 wt.% MgO (maximum value of 311.61 MPa, corresponding to the XL8 sample). For the samples formed by CUP and sintered at 1650 °C (XL12 sample), the maximum CCS value was 318.72 MPa, which corresponds to an improvement of 80.87% with respect to the reference sample of 100 wt.% MgO. Finally, the highest cold crushing strength was reached by the sample XL16, which was formed by CUP + CIP and sintered at 1650 °C, 323.78 MPa, corresponding to an improvement of 80.87% with respect to the reference sample of 100 wt.% MgO.

As it is known, the temperature is the main factor in a ceramic system that leads to a correct sintering process, thus increasing the mechanical properties among other properties, as was corroborated in these refractory systems. However, at low sintering temperatures (1550 °C), it was confirmed that the forming method controls the increase in mechanical resistance.

In these samples, three mechanisms were present as the concentration of ZrO_2_ increased: (i) a crystalline structure change mechanism with energy absorption, i.e., at a high concentration of ZrO_2_ nanoparticles, more particles undergo a structural change, which helps to stop the crack propagation. When a crack approaches a ZrO_2_ particle (tetragonal zirconia), it is transformed into a new crystalline structure (monoclinic structure) by displacement transformation. This structure change implies energy absorption, causing the crack to slow down and stop [48].

This transformation is not only induced by temperature change but also occurred by a diffusionless shear process at near sonic velocities [3,49,50]. (ii) A volumetric expansion mechanism is believed to be another factor that helps stop cracks. A volumetric expansion (3–5%) is generated by the ZrO_2_ phase transition from tetragonal to monoclinic. This transition occurs by displacement transformation induced by pressure [3,51,52,53,54]. The expansion is carried out in the periphery of the ZrO_2_ particles. Since these particles increase in volume, they help to deflect or break the cracks [3]. (iii) With an increase in ZrO_2_ nanoparticles content and with the CUP + CIP method, samples densified with homogenous pore dispersion into the MgO matrix (isolated pores) were obtained, helping to increase the mechanical resistance.

### 3.4. Chemical States Analyses

An XPS analysis was carried out in this work to corroborate the results obtained in XRD analysis. According to the best results in previous analyses, the samples pressed by CUP + CIP (XL5, XL8, XL13, and XL16 samples) were studied by XPS high-resolution spectra after the sintering process (Figure 6). Figure 6a–f corresponds to peaks of the binding energy of Mg 1 s, Ca 2 p, and O 1 s for MgO samples sintered at 1550 °C and 1650 °C, respectively. Figure 6g shows high-resolution spectra of zirconia nanoparticles powders before being incorporated into the matrix (MgO). In this case, two peaks can be observed, which are accredited to Zr 3 d components of spin-orbit splitting (three d_5/2_ and three d_3/2_ orbitals), with binding energies of 181.92 eV and 184.29 eV, respectively. The energy difference ΔE, between the Zr 3 d doublets, corresponds to 2.4 eV [55]. The image inserted in Figure 6g shows the high-resolution O 1 s peaks in the binding energies at 529.85 eV [56] and 531.02 eV [57,58], corresponding to ZrO_2_ nanoparticles and the carbon tape, respectively. Figure 6h–o corresponds to high-resolution XPS of Mg 1 s, Ca 2 p, O 1 s, and Zr 3 d for samples with 5 wt.% of ZrO_2_ nanoparticles sintered at 1550 °C and 1650 °C, respectively.

The deconvoluted core-level spectrum of Figure 6a,d,h,l corresponds to peaks of the binding energy of Mg 1 s of the MgO and Mg(OH)_2_ phases. The corresponding binding energy for MgO (red line) was 1303.76, 1304.01, 1303.99, and 1303.56 eV. Meanwhile, the binding energy of Mg(OH)_2_ (blue line) was detected at 1302.99, 1303.02, 1302.81, and 1302.90 eV. This phase was detected due to a reaction on the sample surface with the environment humidity.

The deconvoluted core-level spectrum of Figure 6b,e,i,m corresponds to peaks of the binding energy of Ca 2 p of the CaO phase, identified by green lines. In the samples with ZrO_2_ nanoparticles at 1550 °C and 1650 °C (Figure 6i,m), a chemical shift towards lower binding energies was evident in comparison with those obtained in samples of MgO (Figure 6b,e) sintered at 1550 °C and 1650 °C, respectively. The above is attributed to the formation of the CaZrO_3_ phase with binding energies at 347.20 (2 p_3/2_) and 350.81 (2 p_1/2_) at 1550 °C, and 347.25 (2 p_3/2_) and 350.88 (2 p_1/2_) eV at 1650 °C. These results agree with binding energies values reported in other works [59,60]. XPS analysis confirmed the presence of CaZrO_3_ formed in the sample when the ZrO_2_ nanoparticles were added, as was observed in XRD analyses [42]. The energy difference (ΔE) was ~3.6 eV and corresponded to Ca 2p doublets, which agrees with the literature [60]. The CaO is an impurity in the raw material that reacts with the ZrO_2_ to form CaZrO_3_ in MgO samples containing ZrO_2_ nanoparticles [3,47].

Figure 6c,f,j,n shows the deconvolution of binding energies of O 1 s for MgO (Figure 6c,f) and MgO containing 5 wt.% of ZrO_2_ nanoparticles (Figure 6j,n), each sintered at 1550 and 1650 °C. In general, three peaks associated with the phases MgO, Mg(OH)_2_, and CaO phases are observed. However, with ZrO_2_ nanoparticles, two more peaks corresponding to ZrO_2_ and CaZrO_3_ (violet and brown line, respectively) are observed. The different binding energies values of O 1 s eV peaks of MgO, Mg(OH)_2_, CaO, ZrO_2_, and CaZrO_3_ phases are listed in Table 2. Besides, a comparison is made with those values found in the literature.

Figure 6k,o, shows the deconvolution of the Zr 3 d core level spectrum for the sample with 5 wt.% ZrO_2_. A doublet in their high-resolution spectra due to spin-orbital coupling is shown. The binding energies correspond to 181.15 (Zr 3 d_5/2_) and 183.42 eV (Zr 3 d_3/2_) (Figure 6k) for samples sintered at 1550 °C and 181.23 (Zr 3 d_5/2_) and 183.69 eV (Zr 3 d_3/2_) (Figure 6o) for samples sintered at 1650 °C. The energy difference (ΔE) between the peaks is ~2.4 eV, which completely agrees with the raw material analyzed of ZrO_2_ nanoparticles (Figure 6g).

Table 3 presents the XPS binding energies for high-resolution analysis of Mg 1 s, Ca 2 p, and Zr 3 d of each sample analyzed.

In summary, the changes in chemical states in the elements (Mg, Ca, O, Zr) of the analyzed samples were identified by the XPS technique. The major change was detected in Ca, O, and Zr due to the CaZrO_3_ phase formed during the sintering process. According to the literature in Table 2, the bond energies detected in 532.2 and 532.1 eV for oxygen represents the formation of the CaZrO_3_ phase. Meanwhile, the Ca and Zr displayed shifts towards lower bond energies corresponding to a reduction in each element.

### 3.5. Chemical Properties

#### 3.5.1. MgO Sample Tested with Silicomanganese Slag

Figure 7 shows the morphological analysis of the XL13 sample once it was chemically attacked with silicomanganese slag. The XL13 sample corresponds to MgO formed by CUP + CIP and sintered at 1650 °C. Moreover, a graph corresponding to the concentration of silicomanganese slag elements as a function of the penetration distance is plotted. As a guidance, a schematic representation of the cross-cut section of the specimen used for chemical analysis is observed.

Figure 7a shows the SEM micrograph corresponding to the upper area of the sample (lateral wall of the hole) where the slag was placed. Strong adhesion of silicomanganese slag to the MgO grains is observed, as indicated by the crusts shown in the image. Figure 7b shows the interface between magnesia refractory/silicomanganese slag, where the first contact between MgO grains and slag is displayed. A dense microstructure can be observed due to a slag infiltration into the pores.

Figure 7c corresponds to the sample’s lower area, where silicomanganese slag penetration into the matrix can be observed. The slag is located both at the grain boundaries and triple points. The results of the semi-quantitative analysis (EDX) are presented in Figure 7d.

The plotted graph shows the concentration of slag elements associated with the slag (Mn, Ca, Si, and Al) ordered from highest to lowest traces as a function of the penetration distance concerning the height of the sample (6 mm). The letters marked as A (upper area of the sample), B (interface, magnesia refractory/silicomanganese slag), and C (lower area of the sample) and the red dotted lines on the graph correspond to the place (distance from y = 0) where the morphological SEM study of the samples was performed. A concentration of 1.11% of slag elements was detected at 181 µm from the bottom of the sample. Likewise, another significant concentration of 7.26% of slag elements was observed at around 763 µm to the bottom of the sample. The slag concentration was 7.63% at a distance from 1927 µm to 3381 µm to the bottom of the sample. This zone represents the boundary between MgO grains and the bottom of the hole where the silicomanganese slag was placed. The slag concentration decreased (~2.94%) due to a slag penetration towards the bottom and walls of the refractory at the zone covered by the hole (3490 µm to 6000 µm distance). Figure 7e shows the cross-section sample and the zones where the SEM micrograph analyses were made (indicated by letters A, B, and C). The path followed to perform the chemical microanalysis by EDX is indicated by the dotted red line.

#### 3.5.2. MgO Containing 5 wt.% of ZrO_2_ Nanoparticles Tested with Silicomanganese Slag

Figure 8 shows the morphological analysis of the sample XL16 chemically attacked with silicomanganese slag. The sample XL16 corresponds to the MgO sample that contains 5 wt.% of ZrO_2_ nanoparticles formed by CUP + CIP and sintered at 1650 °C. The graph corresponding to the concentration of silicomanganese slag elements as a function of the penetration distance is also plotted. A schematic representation of the cross-cut section of the specimen used for chemical analysis is observed.

In Figure 8a, the wall of the hole where the silicomanganese slag powders were placed to carry out the chemical corrosion analysis is observed. The image shows the molten slag strongly adhered to the MgO and ZrO_2_ grains. In some cases, a strong slag adherence leads to the development of a crust that protects the refractory to direct contact with the melted slag, gas, and fluxes. Figure 8b shows a micrograph corresponding to the area below the interface refractory/slag, i.e., the lowest area where the slag was deposited. Slag penetration was mainly observed at the grain boundaries of magnesia. However, some slag penetrated closed pores inside the MgO grains, as was observed in the micrograph. Some agglomerates composed of ZrO_2_ nanoparticles are surrounded by some elements from the slag (Mn, Ca, Al, Si, Ti). ZrO_2_ nanoparticles might be a barrier at both triple points and grain boundaries to avoid the advance of the SiMn slag in the refractory. As was analyzed, no CaZrO_3_ traces were found.

This phenomenon can be attributed to a possible interaction between CaZrO_3_ and the slag. This reaction might change the slag viscosity that prevents slag penetration, i.e., a higher viscosity will help to control or stop the slag infiltration, as reported in other CaZrO_3_ refractory systems [23].

The formation of CaZrO_3_ was not the objective of the present research work. However, magnesia raw material included CaO among its impurities (up to 1 wt.%), which by reaction with the zirconia nanoparticles lead to the CaZrO_3_ formation. CaZrO_3_ is located at the grain junctions and triple points since ZrO_2_ nanoparticles react with CaO forming CaZrO_3_ at these points, also promoted by the utilization of fine particles. The reaction of CaZrO_3_ formation is widely reported in the literature [71,72,73]:

On the other hand, calcium zirconate, due to its beneficial properties, such as high melting point (2345 °C), low linear thermal expansion coefficient (11.05·10^–6^ K^−1^), high hardness (9.5 GPa), low thermal conductivity, high chemical stability, and corrosion resistance to alkalis (especially KOH), alkaline earth metals, slags [74,75,76] and other environments such as NaVO_2_–Na_2_SO_4_ mixture [77], can be applied in numerous fields of industry. This chemical compound is stable up to the temperature of 1800 °C, at which the transformation from orthorhombic to cubic phase takes place [78]. CaZrO_3_ is also a promising candidate for use in titanium metallurgy. Meanwhile, CaZrO_3_ is considered a material for refractory linings in cement kilns when is added to the MgO refractories [28,79,80]. CaZrO_3_ was successfully verified as a competitive refractory material for the melting process of titanium and the production of cement clinker. However, only a few research works concerned the interaction of this refractory compound with steelmaking and ironmaking slags.

Although CaZrO_3_ is stable at 1550 °C in contact with the slag, it is suggested in the present study (considering the chemical composition of the silicomanganese slag (wt.%): 26.88% SiO_2_, 24.85% CaO, 24.56% MnO, 12.41% Al_2_O_3_, 4.55% MgO, 1.21% BaO, 0.76% Na_2_O, 0.80 K_2_O, and balanced others (Fe, P, Ti, Sr)) that the following products can be formed by reaction with the metallurgical slag: calcium zirconate (CaZrO_3_), zirconium oxide stabilized with Ca^2+^ (ZrO_2_stab., Zstab.), gehlenite (C_2_AS), and baghdadite (Ca_3_Zr[O_2_|Si_2_O_7_], C_3_ZS_2_). These phases coexist in the four-component diagram A–C–Z–S (Al_2_O_3_–CaO–ZrO_2_–SiO_2_) [81]. The suggested reaction is attributed to the exposure time in contact with slag and the main chemical compounds that constituted the silicomanganese slags.

In Figure 8c, a slag-free microstructure is observed since, in the EDX analysis, only MgO and ZrO_2_ are found.

Figure 8d shows the results of the semi-quantitative analysis by the EDX technique. The plotted graph shows the concentration of slag elements (Mn, Ca, Al, Si, and Ti) ordered from highest to lowest traces as a function of the penetration distance concerning the height of the sample (6 mm). The letters (A, B, and C) and red dotted lines on the graph correspond to the area where the morphological study was carried out on the sample. The analysis allowed us to observe that there was no slag penetration at the bottom of the refractory specimen. A slag concentration of 0.85% was detected above 1636 µm concerning the bottom of the sample.

This concentration might result from the slag infiltration into the residual open pores after the sintering process. The slag concentration was 3.22% at a distance from 1945 µm to 3109 µm with respect to the bottom of the sample. This concentration might be due to the slag infiltration through the MgO grain boundaries. At the interface between magnesia refractory/silicomanganese slag (~3467 µm concerning the bottom of the sample), the concentration of slag elements was 4.32%. The first contact and infiltration of the slag took place in this zone of the refractory. The slag concentration was ~11.5% in the wall of the hole where slag was deposited (at a distance from 3500 µm to 6000 µm). This value was higher concerning the sample of 100 wt.% of MgO due to a higher dense refractory matrix that retained more slag elements in this zone. Figure 8e shows the cross-section sample and the zones where the SEM micrograph analyses were made, indicated by letters (A, B, and C). The path followed to perform the chemical microanalysis by EDX is indicated by the dotted red line.

In summary, the change in slag viscosity (a higher slag viscosity) due to the reaction between CaZrO_3_ and the melted slag is the main mechanism that leads to a decrease in corrosion attack. Moreover, the addition of ZrO_2_ nanoparticles represents a barrier at triple points and grain boundaries to avoid the advance of the slag in the refractory. These two mechanisms help to control or stop the slag penetration.

## 4. Conclusions

The behavior of MgO ceramics reinforced with ZrO_2_ nanoparticles against SiMn slag was studied in this manuscript. Different contents of ZrO_2_ nanoparticles (0, 1, 3, and 5 wt.%) were considered, as well as different methods of obtaining green compacts (CUP and CUP + CIP) and sintering temperatures (1550 °C and 1650 °C).

Three mechanisms promote the densification of the MgO microstructure through the addition of ZrO_2_ nanoparticles: (i) CaZrO_3_ phase formation (formed due to in situ reaction between CaO (1 wt.%) and the ZrO_2_ nanoparticles) with a higher density (4.95 g/cm^3^) than the MgO matrix (3.58 g/cm^3^); (ii) development of a ceramic bonding through the CaZrO_3_ formation; (iii) pinning effect due to the specific location of ZrO_2_ and CaZrO_3_ at the grain boundaries and triple points. The lowest values of porosity (14.485%) and the highest value of density (3.0599 g/cm^3^) were measured in the samples with 5 wt.% of ZrO_2_ nanoparticles formed by CUP + CIP and sintered at 1650 °C.

The main change by the XPS technique was detected in Ca, O, and Zr due to the CaZrO_3_ phase formed during the sintering process. The bond energies detected in 532.2 and 532.1 eV for oxygen correspond to the formation of the CaZrO_3_ phase.

The increase in mechanical resistance for samples with ZrO_2_ nanoparticles is a consequence of this phase. The maximum value of CCS is observed when 5% ZrO_2_ nanoparticles are added. Samples were obtained by CUP + CIP and sintered at 1650 °C. The value of cold crushing strength is 80.87% greater than the reference sample with only MgO.

The corrosion resistance is controlled by the CaZrO_3_ but also by the decreased porosity, which difficulties the advance of the slag in the refractory. Similarly, ZrO_2_ nanoparticles act as a barrier both at the triple points and the grain boundaries, avoiding the advance of silicomanganese slag in the magnesia refractory.

## Figures and Tables

**Figure 1 materials-15-02421-f001:**
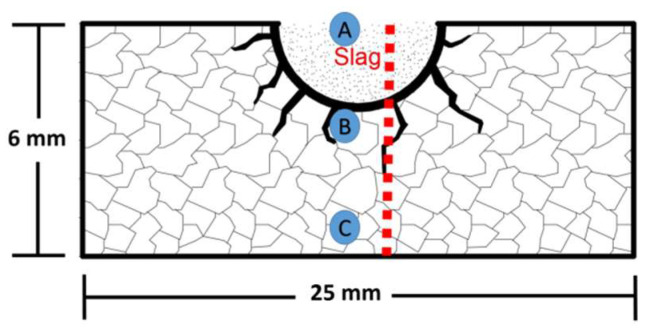
Scheme representing the chemical resistance test of silicomanganese slag on refractory samples where: A = slag zone, B = slag/refractory interface, and C = refractory zone.

**Figure 2 materials-15-02421-f002:**
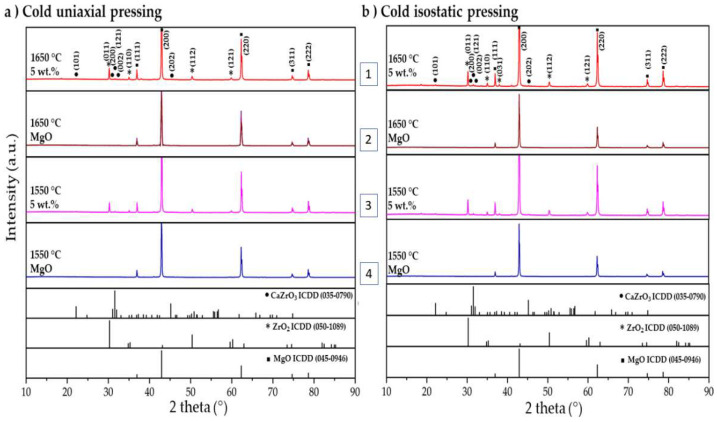
XRD pattern of the sample 100 wt.% MgO (brown and blue line, 2 and 4) and MgO sample containing 5 wt.% of ZrO_2_ nanoparticles (red and pink line, 1 and 3) sintered at 1550 °C and 1650 °C, pressed by: (**a**) Cold uniaxial pressing (CUP) and (**b**) cold isostatic pressing after cold uniaxial pressing (CUP + CIP).

**Figure 3 materials-15-02421-f003:**
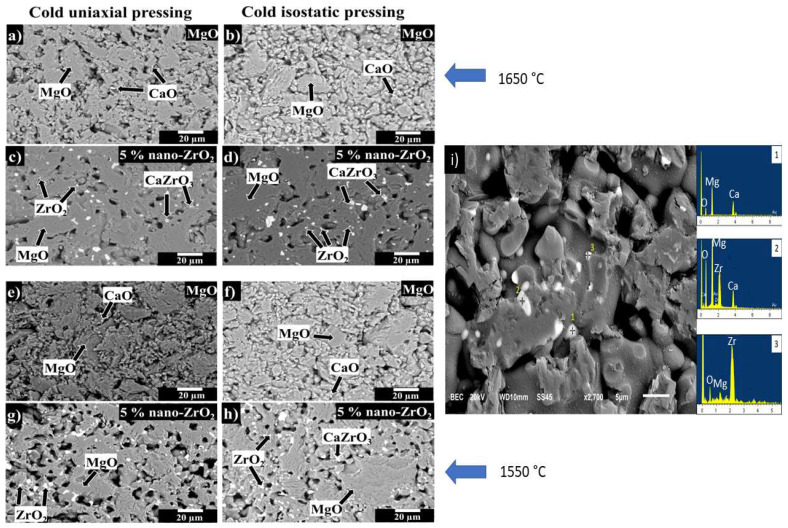
SEM images. (**a**,**b**) 100 wt.% MgO samples and (**c**,**d**) specimens MgO containing 5 wt.% of ZrO_2_ nanoparticles, all sintered at 1650 °C. (**e**,**f**) 100 wt.% MgO and (**g**,**h**) MgO containing 5 wt.% of ZrO_2_ nanoparticles, all sintered at 1550 °C. (**i**) shows the shape of each of the phases identified as point 1 (CaO), point 2 (CaZrO_3_), point 3 (ZrO_2_). (**a**,**c**,**e**,**g**) correspond to samples pressed by CUP. (**b**,**d**,**f**,**h**) correspond to samples pressed by a CUP + CIP.

**Figure 4 materials-15-02421-f004:**
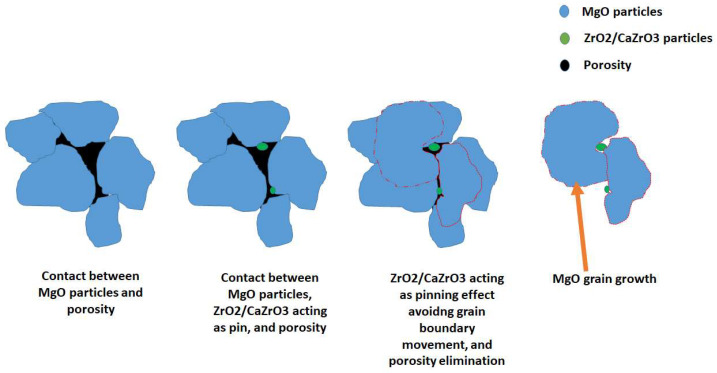
Influence of ZrO_2_ nanoparticles on the densification of the samples.

**Figure 5 materials-15-02421-f005:**
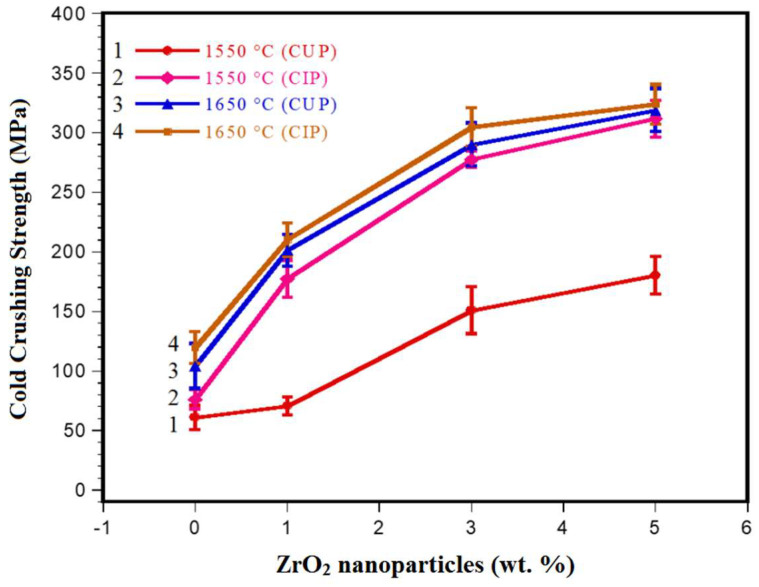
Variations in the cold crushing strength (CCS) of magnesia refractories with different contents of ZrO_2_ nanoparticles formed by CUP or CUP + CIP and sintered at 1550 °C and 1650 °C.

**Figure 6 materials-15-02421-f006:**
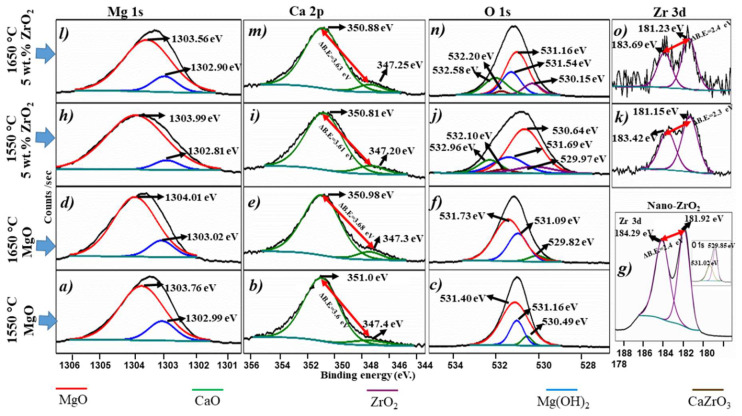
XPS high-resolution spectra of (**a**) Mg 1 s, (**b**) Ca 2 p, and (**c**) O 1 s of MgO sintered a 1550 °C, (**d**) Mg 1 s, (**e**) Ca 2 p and (**f**) O 1 s of MgO sintered a 1650 °C. Spectra in (**g**) correspond to Zr 3 d for nanoparticles of ZrO_2_. The image in the inset in (**g**) shows two peaks; one corresponds to the O 1 s spectra of nano-ZrO_2_ (higher peak) and the other to C 1 s corresponding to carbon tape. (**h**–**o**) Spectra of Mg 1 s, Ca 2 p, O 1 s, and Zr 3 d correspond to samples with 5 wt.% of ZrO_2_ nanoparticles sintered at 1550 °C and 1650 °C, respectively.

**Figure 7 materials-15-02421-f007:**
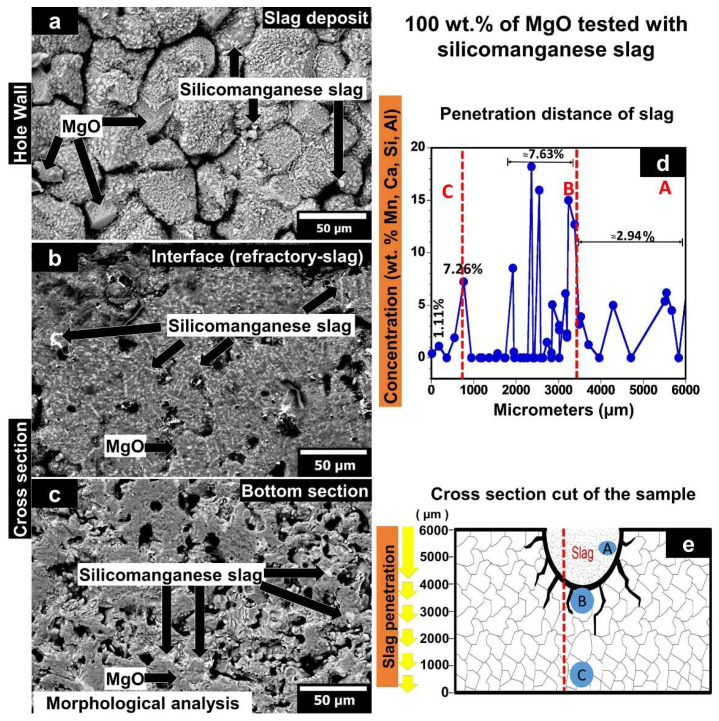
(**a**–**c**) SEM images of 100 wt.% MgO sample sintered at 1650 °C and chemically attacked with silicomanganese slag. (**d**) The concentration of silicomanganese slag elements as a function of the penetration distance. (**e**) Schematic representation of the cross-cut section of the sample used for chemical analysis.

**Figure 8 materials-15-02421-f008:**
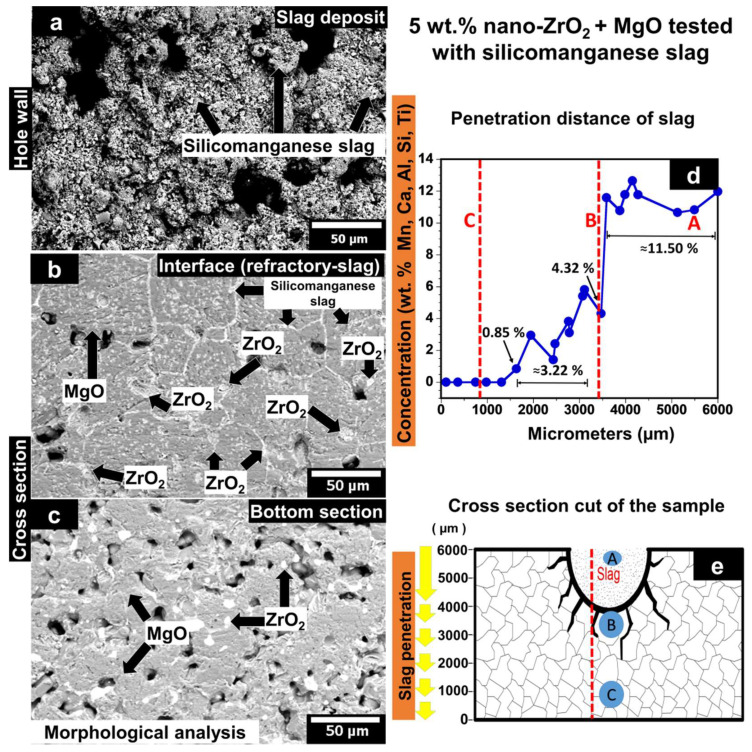
(**a**–**c**) SEM images of the MgO sample that contains 5 wt.% of ZrO_2_ nanoparticles sintered at 1650 °C and chemically attacked with silicomanganese slag. (**d**) The concentration of silicomanganese slag elements as a function of the penetration distance. (**e**) Schematic representation of the cross-cut section of the sample used for chemical analysis.

**Table 1 materials-15-02421-t001:** Sintering temperature, pressing method (cold uniaxial pressing (CUP) or cold uniaxial pressing (CUP) + Cold Isostatic Pressing (CIP)), batch composition and physical properties of samples analyzed.

Sintering Temperature	Pressing Method	Batch Composition			Physical Properties	
		Simple Code	ZrO_2_ Nanoparticles (wt.%)	MgO (wt.%)	Density (g/cm^3^)	Porosity (%)
1550 °C	CUP Batch 1	XL1	0	100	2.62	28.46
		XL2	1	99	2.66	27.21
		XL3	3	97	2.88	20.83
		XL4	5	95	2.89	20.90
1550 °C	CUP + CIP Batch 3	XL5	0	100	2.67	27.03
		XL6	1	99	2.67	26.79
		XL7	3	97	2.90	19.69
		XL8	5	95	2.90	20.24
1650 °C	CUP Batch 2	XL9	0	100	2.65	27.54
		XL10	1	99	2.83	21.73
		XL11	3	97	2.95	18.27
		XL12	5	95	3.01	17.22
1650 °C	CUP + CIP Batch 4	XL13	0	100	2.71	26.25
		XL14	1	99	2.88	20.57
		XL15	3	97	2.99	16.43
		XL16	5	95	3.06	14.49

**Table 2 materials-15-02421-t002:** Binding energy corresponds to O 1 s eV peaks of MgO, Mg(OH)_2_, ZrO_2_, CaO, and CaZrO_3_ phases for samples of 100 wt.% MgO sintered a 1550 °C (XL5) and 1650 °C (XL13), samples of MgO containing 5 wt.% of ZrO_2_ nanoparticles sintered at 1550 °C (XL8) and 1650 °C (XL16).

Sample	O 1 s eV				
	MgO	Mg (OH)_2_	ZrO_2_	CaO	CaZrO_3_
ZrO_2_ nanoparticles			529.85		
XL5	531.4	531.16		530.49	
XL8	530.64	531.69	529.97	532.96	532.1
XL13	531.73	531.09		529.82	
XL16	531.16	531.54	530.15	532.58	532.2
From the literature	529.2 [58], 530.4 [57,58], 531.2 [61].	531.15 [58], 531.5 [58], 531.6 [61], 532.5 [58],	527.1 [62], 529.7 [63], 530.0 [63].	530.10 [64], 532.5 [59].	531.9 [65], 532.2 [59]

**Table 3 materials-15-02421-t003:** Binding energy corresponds to Mg 1 s, Ca 2 p, and Zr 3 d of MgO, Mg(OH)_2_, CaO, CaZrO_3_, and ZrO_2_ phases, corresponding to samples of 100 wt.% MgO sintered a 1550 °C (XL5) and 1650 °C (XL13), MgO containing 5 wt.% of ZrO_2_ nanoparticles sintered at 1550 °C (XL8) and 1650 °C (XL16).

Sample	Mg 1 s eV		Ca 2 p eV		Zr 3 d eV	
	MgO	Mg(OH)_2_	CaO 2p_1/2_	CaO 2p_3/2_	ZrO_2_ 3d_3/2_	ZrO_2_ 3d_5/2_
ZrO_2_ nanoparticles					184.29	181.92
XL5	1303.76	1302.99	351.00	347.40		
XL8	1303.99	1302.81	350.81	347.20	183.42	181.15
XL13	1304.01	1303.02	350.98	347.30		
XL16	1303.56	1302.90	350.88	347.25	183.69	181.23
From the literature	1303.8 [66], 1303.9 [57,58], 1303.4 [67].	1301.1 [57], 1301.98 [58], 1302.2 [66], 1302.7 [68].	348 [59], 349.7 [64]. 351.0 [59,60].	346.5 [59], 347.7 [59].	183 [59], 184.9 [69].	181.1 [70], 182 [63].

## Data Availability

Not applicable.

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
