# Peer review of "MgO–ZrO2 Ceramic Composites for Silicomanganese Production"

_materials, 2022, doi:10.3390/ma15072421_

Round 1

Reviewer 1 Report

In this manuscript (materials-1655639), the behavior of MgO-ZrO2 ceramic composites with different ZrO2 nanoparticle contents in presence of silicomanganese slags is studied, and the ceramic composite against the silicomanganese slag was characterized by XPS, XRD and SEM-EDX. It showed that corrosion is controlled by the change in slag viscosity due to the reaction between CaZrO3 and the melted slag, and the addition of ZrO2 nanoparticles can extend the life of furnaces used to produce silicomanganese.  The results are meaningful to provide some insights into refratories. Hence, I recommend that this manuscript be published on Materials after the authors revise the following questions:

  1. Why only fine MgO powders were used for the preparation of the sample, usually MgO grain with size of 1-5 mm and fine powder used for refractories.
  2. Which crystalline type of ZrO2 used in present paper, and why?
  3. Which reaction between CaZrO3 and the melted slag occurred during the experiment, please explain in detail.

Author Response

Dear Reviewer 1, thank you for your insightful comments about the manuscript. We answer your comments below the answer.

In this manuscript (materials-1655639), the behavior of MgO-ZrO2 ceramic composites with different ZrO2 nanoparticle contents in presence of silicomanganese slags is studied, and the ceramic composite against the silicomanganese slag was characterized by XPS, XRD and SEM-EDX. It showed that corrosion is controlled by the change in slag viscosity due to the reaction between CaZrO3 and the melted slag, and the addition of ZrO2 nanoparticles can extend the life of furnaces used to produce silicomanganese.  The results are meaningful to provide some insights into refratories. 

Hence, I recommend that this manuscript be published on Materials after the authors revise the following questions:

  1. Why only fine MgO powders were used for the preparation of the sample, usually MgO grain with size of 1-5 mm and fine powder used for refractories.

Thank you for your comment, now the explanation was added from line 92 to line 105. We agree with you. We know that aggregates comprise raw materials larger than about 5 mm and constitute, by weight, about 70% of a refractory product. However, our study is focused on the matrix and the bonding elements and is explained as follows:

It is well known that the fine fraction is considered the weakest constituent of a refractory matrix. Therefore, it has to be reinforced by a strong bond development. The bonding strength represents one of the main microstructural characteristics contributing to the reliable refractory matrix establishment. Increasing the bonding strength, the resistance against many kinds of stresses during the performance, and structural spalling would be improved. That is why we focused on the fine fraction study since it is the one that has the highest reactivity in a refractory system. Besides, the aggregates are the main refractory constituent that supports the mechanical, thermal, and chemical changes in a system. Therefore, we did not consider the aggregates for this study.

On the other hand, there are many studies focused on the fine elements of a refractory (matrix and bonding structure) as was cited below. Through these studies, it has been found that reducing the particle size (<45 µm) helps the thermal sintering process by improving the morphology and microstructure of sintered composites with a beneficial impact on the mechanical, physical and chemical properties [17, 28, 30, 34].

[17]  Gómez, C.; Castillo, G. A.; Rodríguez, E. A.; Vázquez-Rodríguez, F. J.; López-Perales, J. F.; Aguilar-Martínez, J. A.; Fernández-González, D.; García-Quiñonez, L. V.; Das-Roy, T. K.; Verdeja, L. F. Development of an ultra-low carbon MgO refractory doped with α-Al2O3 nanoparticles for the steelmaking Industry: A microstructural and thermo-mechanical study. Materials 2020, 13, 715. https://doi.org/10.3390/ma13030715

[28]  Serena, S.; Sainz, M. A.; Caballero, A. Corrosion behavior of MgO/CaZrO3 refractory matrix by Clinker. J. Eur. Ceram. Soc. 2004, 24, 2399-2406. https://doi.org/10.1016/j.jeurceramsoc.2003.07.007

[30] Gómez-Rodríguez, C.; Das, T. K.; Shaji, S.; Castillo-Rodríguez, G. A.; García-Quiñonez, L.; Rodríguez, E.; González, J. O.;  Aguilar-Martínez, J. A. Effect of addition of Al2O3 and Fe2O3 nanoparticles on the microstructural and physico-chemical evolution of dense magnesia composite. Ceram. Int. 2015, 41, 7751-7758. https://doi.org/10.1016/j.ceramint.2015.02.107

[34]  Ghasemi-Kahrizsangi, S.; Sedeh, M. B.; Dehsheikh, H. G.; Shahraki, A.; Farooghi, M. Densification and properties of ZrO2 nanoparticles added magnesia-doloma refractories. Ceram. Int. 2016, 42, 15658-15663. https://doi.org/10.1016/j.ceramint.2016.07.021

  1. Which crystalline type of ZrO2 used in present paper, and why?

ZrO2 nanoparticles used in this manuscript had monoclinic structure. Zirconium oxide (ZrO2) is a phase that characterizes by exhibiting three polymorphic transformations. It is monoclinic from room temperature up to 1170 ºC, tetragonal from 1171 ºC to 2370 ºC and cubic above 2370 ºC and until the melting point (2715 ºC).

It is well known that there are several phases that stabilize the ZrO2 as the MgO, CaO, Y2O3, which are adequate to stabilize a single crystalline structure of ZrO2 that is to say, that would remain with a certain crystalline structure form room temperature to the melting point (without polymorphic changes (monoclinic-tetragonal-cubic, or vice versa) and, therefore, volume changes that could involve cracks in the parts). As well, we have observed in this research that the ZrO2 particles were completely surrounded by MgO particles (matrix constituent), in agreement with the MgO-ZrO2 binary diagram (the ZrO2 tetragonal phase admits 10% of MgO and the cubic phase admits up to 27% MgO in solid solution). This way, when the composites were sintered at 1600 ºC, there was a change from monoclinic to tetragonal, and this crystallographic change was maintained down to the room temperature as the MgO acted as stabilizer. This can be checked in the X-ray diffraction results in Figure 2 as with 5 wt. % of ZrO2, peaks corresponding to the tetragonal ZrO2 were observed (ICDD 50-1089).

  1. Which reaction between CaZrO3 and the melted slag occurred during the experiment, please explain in detail.

Thank you for your observation. A wide explanation is detailed from lines 484 a 517

The formation of CaZrO3 was not the objective of the present research work. However, the magnesia included CaO among its impurities (up to 1 wt. %), which by reaction with the zirconia nanoparticles lead to the CaZrO3 formation. CaZrO3 is located at the grain junctions and triple points since ZrO2 nanoparticles (located in grain junctions and triple points) react with CaO forming CaZrO3, also promoted by the utilization of fine particles. The reaction taking place to CaZrO3 formation is widely reported in the literature [71-73]:

On the other hand, calcium zirconate due to its beneficial properties, such as high melting point (2345°C), low linear thermal expansion coefficient (11.05·10-6 K-1), high hardness (9.5 GPa), low thermal conductivity, high chemical stability, and corrosion resistance to alkalis (especially KOH), alkaline earth metals, slags [74-76]] and other environments such as NaVO2 - Na2SO4 mixture [77] can be applied in numerous fields of industry. This chemical compound is stable up to the temperature of 1800°C, at which the transformation from orthorhombic to cubic phase takes place [78]. CaZrO3 is also a promising candidate for use in titanium metallurgy. Meanwhile, CaZrO3 is considered a material for refractory linings in cement kilns when is added to the MgO refractories [79-81]. CaZrO3 has been successfully verified as a competitive refractory material for the melting process of titanium and the production of cement clinker. However, only a few research works concerned the interaction of this refractory compound with steelmaking and ironmaking slags.

Although CaZrO3 is stable at 1550°C in contact with the slag, we suggest in the present study the following reaction taking into account the chemical composition of silicomanganese slag from the metallurgical process (wt. %): 26.88% SiO2, 24.85% CaO, 24.56% MnO, 12.41% Al2O3, 4.55% MgO, 1.21% BaO, 0.76% Na2O, 0.80 K2O, balance others (Fe, P, Ti, Sr): Calcium zirconate (CaZrO3), zirconium oxide stabilized with Ca2+ (ZrO2stab., Zstab.), gehlenite (C2AS), and baghdadite (Ca3Zr[O2|Si2O7], C3ZS2). These phases coexist in the four-component diagram A-C-Z-S (Al2O3-CaO-ZrO2-SiO2) [82]. The suggested reaction is attributed to the exposure time in contact with slag and the main chemical compounds that constituted the silicomanganese slags.

  1. Angers, R.; Tremblay, A.; Chaklader, A. C. D., 1972: Formation of CaZrO3 by Solid-State Reaction Between CaO and ZrO2, Journal of the American Ceramic Society, 55 (8), 425-425. https://doi.org/10.1111/j.1151-2916.1972.tb11327.x
  2. Serena, S., 2002: Modelización Termodinámica y cálculo del diagrama de equilibrio de fases ZrO2-CaO-MgO: aplicación al diseño y obtención de materiales de MgO-CaZrO3, Ph. D. Thesis (supervisors: Sáinz, M. A; Vázquez, B. A.). Universidad Autónoma de Madrid, Facultad de Ciencias and Instituto de Cerámica y Vidrio (C.S.I.C.). Date: 21-03-2002. http://hdl.handle.net/10486/673998
  3. Baudín de la Lastra, C.; Pena, P.; Obregón, A.; Rodríguez-Galicia, J. L. 2010: Mechanical Behaviour of MgO-CaZrO3-based Refractories for Cement Kilns, Advances in Science and Technology, 70, 47-52. https://doi.org/10.4028/www.scientific.net/AST.70.47
  4. Schafföner, C. G. Aneziris, H. Berek, J. Hubálková, A. Priese, Fused calcium zirconate for refractory applications, J. Eur. Ceram. Soc. 33 (2013) 3411–3418. doi:10.1016/j.jeurceramsoc.2013.07.008
  5. F. Hou, Ab initio calculations of elastic modulus and electronic structures of cubic CaZrO3, Phys. B Condens. Matter. 403, 17 (2008) 2624–2628. doi: 10.1016/j.physb.2008.01.025
  6. Szczerba, Modyfikowane magnezjowe materiały ogniotrwałe, Polskie Towarzystwo Ceramiczne,
    Kraków, 2007
  7. Silva, F. Booth, L. Garrido, E. Aglietti, P. Pena, C. Baudín, Sliding wear of CaZrO3-MgO composites against ZrO2 and steel, J. Eur. Ceram. Soc. 37, 1 (2017) 297–303. doi:10.1016/j.jeurceramsoc.2016.07.029
  8. Szczerba, Wyroby magnezjowe z cyrkonianem wapnia w osnowie, Mater. Ceram./Ceramic Mater. 61, 1 (2009) 7–11
  9. F. Lang, J. G. You, X. F. Zhang, X. D. Luo, S. Y. Zheng, Effect of MgO on thermal shock resistance of CaZrO3 ceramic, Ceram. Int. 44, 18 (2018) 22176–22180. doi: 10.1016/j.ceramint.2018.08.333
  10. Serena, M. A. Sainz, A. Caballero, Corrosion behavior of MgO/CaZrO3 refractory matrix by clinker, J. Eur. Ceram. Soc. 24, 8 (2004) 2399–2406, 2004, doi: 10.1016/j.jeurceramsoc.2003.07.007
  11. M. M. Ewais, I. M. I. Bayoumi, Fabrication of MgO-CaZrO3 refractory composites from Egyptian dolomite as a clinker to rotary cement kiln lining, Ceram. Int. 44, 8 (2018) 9236–9246. doi: 10.1016/j.ceramint.2018.02.134
  12. Pena, S. De Aza, Compatibility relations of Al2O3 in the system ZrO2-Al2O3-SiO2-CaO, J. Am. Ceram. Soc. 67, 1 (C3-C5) 1984

Reviewer 2 Report

In this study, MgO-ZrO2 ceramics with different amount of ZrO2 were fabricated at different sintering temperatures, and the microstructures, crushing strength and chemical properties of these ceramics were also investigated. The authors found that adding ZrO2 into MgO can extend the life of furnaces used to produce silicomanganese. However, there are some mandatory improvements for the paper before acceptance.

  1. In Materials Section 2.1 , the authors gave the fraction of each composition for MgO raw material. Is the fraction a mass fraction? Please specify it in the paper.
  2. The authors proposed that t→m phase transition of ZrO2 occurred during fracture. So, there are two issues should be clarified in the revised paper. Firstly, the lattice structure is tetragonal or monoclinic for the raw material of ZrO2? Secondly, the lattice structure of ZrO2 in the fractured MgO-ZrO2 ceramics should be identified.
  3. The specimen dimensions for crushing strength testing should be given in the 2.3.3 Section.
  4. In Fig.2, the diffraction peaks of (011) and (110) for ZrO2 in XL4 sample were not detected by XRD, which is unlike in other samples. Why?
  5. In Fig.3, the shape and color of CaO, ZrO2 and CaZrO3 phases are not clear enough. For example, the authors wrote “CaO have an angular morphology”. But, it is difficult to identify the shape of CaO from Fig.3. So, the high magnification microstructures should be added in the paper.
  6. In Fig.3g, no CaO/CaZrO3 phase was detected by SEM in XL4 ? Why?
  7. The authors indicated that the pinning effect of ZrO2 and CaZrO3 particles reduced porosity and increased density of the MgO-ZrO2 ceramics. Please explain the reason of the pinning effect on porosity and density. Moreover, the grain size of MgO with ZrO2 is higher than that of MgO without ZrO2.
  8. In Section 3.3, the authors wrote “Fig.4 shows the mechanical behavior of ...”. “mechanical behavior” is not appropriate here.
  9. In Fig.6(d) and Fig.7(d), the concentration of slag element was detected by spot analysis? If so, the elemental concentration may be seriously affected by spot size and location(at grain boundary or in grain).
  10. In Conclusion Section, the three mechanisms for enhanced strength should not be listed in conclusion. Firstly, t→m phase transition toughening mechanism was well known. More importantly, whether t→m phase transition occurred during strength testing has not been confirmed in this study.

Author Response

Dear Reviewer, thank you for your insightful comments about the manuscript. We answer your comments below the answer.

In this study, MgO-ZrO2 ceramics with different amount of ZrO2 were fabricated at different sintering temperatures, and the microstructures, crushing strength and chemical properties of these ceramics were also investigated. The authors found that adding ZrO2 into MgO can extend the life of furnaces used to produce silicomanganese. However, there are some mandatory improvements for the paper before acceptance.

  1. In Materials Section 2.1, the authors gave the fraction of each composition for MgO raw material. Is the fraction a mass fraction? Please specify it in the paper.

The fraction corresponds to weight percentage (wt. %). We have added this information in lines 129 and 130 and we have also modified the section 2.2 to include that:

Weight percentages of ZrO2 nanoparticles were added to magnesia powders considering the following relation: (100-X) wt. % MgO + X wt. % of ZrO2, where X = 0, 1, 3 and 5.

  1. The authors proposed that t→m phase transition of ZrO2 occurred during fracture. So, there are two issues should be clarified in the revised paper. Firstly, the lattice structure is tetragonal or monoclinic for the raw material of ZrO2? Secondly, the lattice structure of ZrO2 in the fractured MgO-ZrO2 ceramics should be identified.

This information is explained in the article in lines 109-116 and lines 227-235

First:

ZrO2 nanoparticles used in this manuscript had monoclinic structure. Zirconium oxide (ZrO2) is a phase that characterizes by exhibiting three polymorphic transformations. It is monoclinic from room temperature up to 1170 ºC, tetragonal from 1171 ºC to 2370 ºC and cubic above 2370 ºC and until the melting point (2715 ºC).

It is well known that there are several phases that stabilize the ZrO2 as the MgO, CaO, Y2O3, which are adequate to stabilize a single crystalline structure of ZrO2 that is to say, that would remain with a certain crystalline structure form room temperature to the melting point (without polymorphic changes (monoclinic-tetragonal-cubic, or vice versa) and, therefore, volume changes that could involve cracks in the parts). As well, we have observed in this research that the ZrO2 particles were completely surrounded by MgO particles (matrix constituent), in agreement with the MgO-ZrO2 binary diagram (the ZrO2 tetragonal phase admits 10% of MgO and the cubic phase admits up to 27% MgO in solid solution). This way, when the composites were sintered at 1600 ºC, there was a change from monoclinic to tetragonal, and this crystallographic change was maintained down to the room temperature as the MgO acted as stabilizer. This can be checked in the X-ray diffraction results in Figure 2 as with 5 wt. % of ZrO2, peaks corresponding to the tetragonal ZrO2 were observed (ICDD 50-1089).

Second:

We think that theoretically there is a structural change of the tetragonal phase to the monoclinic phase because the specimens before breaking maintained the ZrO2 tetragonal phase according to the file ICDD 50-1089 (see Figure 2). Additionally, we know that polymorphic changes are not only induced by temperature change but also occurred by a diffusionless shear process at near sonic velocities [3, 49, 50]. That is to say, there is a structural transformation by displacement induced by pressure [3, 51-54].

According to [48] when a crack approaches a ZrO2 particle (tetragonal zirconia), it is transformed into a new crystalline structure (monoclinic structure) by displacement transformation. This structure change implies energy absorption, causing the crack to slow down and stop [48].

  • Verdeja, L. F.; Sancho, J. P.; Ballester, A.; González, R. Refractory and Ceramic Materials; Síntesis: Madrid, Spain.
  • Ashby, J. M. Engineering Materials 2: An Introduction to Microstructures, Processing and Design; Butterworth-Heinemann, Burlimgton, USA, 2005
  • Stevens, R. Magnesium Elektron, Zirconia and Zirconia Ceramics; Magnesium Elektron, Manchester, United Kingdom, 1986.
  • Wolten, G .M. Diffusionless phase transformations in zirconia and hafnia. Am. Ceram. Soc. 1963, 46, 418-422. https://doi.org/10.1111/j.1151-2916.1963.tb11768.x
  1. The specimen dimensions for crushing strength testing should be given in the 2.3.3 Section.

We have included the dimensions of the specimens in section 2.3.3. in line 170-171. “Specimens of sintered samples corresponding to cylindrical specimens with a diameter of 24 mm and a height of 11 mm”.

  1. In Fig.2, the diffraction peaks of (011) and (110) for ZrO2 in XL4 sample were not detected by XRD, which is unlike in other samples. Why?

Thank you for this comment. There was a mistake with the figure. We have added in line 215, the X-ray diffraction pattern corresponding to the XL4 sample. We have also included the planes corresponding to the ZrO2 phase and some other modifications were also made to make more understandable the graph.

Figure 2. XRD pattern of the sample 100 wt. % MgO (brown and blue line, 2 and 4) and MgO sample containing 5 wt. % of ZrO2 nanoparticles (red and pink line, 1 and 3) sintered at 1550 °C and 1650 °C, pressed by: (a) Cold uniaxial pressing and (b) Cold isostatic pressing.

  1. In Fig.3, the shape and color of CaO, ZrO2 and CaZrO3 phases are not clear enough. For example, the authors wrote “CaO have an angular morphology”. But, it is difficult to identify the shape of CaO from Fig.3. So, the high magnification microstructures should be added in the paper.

We have added in line 243 a new micrograph Figure 3(i), corresponding to the XL4 sample, where it is posible to see the three phases corresponding to the CaO (point 1), CaZrO3 (point 2) and ZrO2 (point 3) and the matrix of MgO.

Now the following paragraph was inserted at line 281 to 285.

Figure 3 (i) shows the shape of the phases detected in the XL4 sample:  Point 1: CaO, it is possible to see that they are bigger than the nanoparticles and they appear in blurred white color. Point 2: CaZrO3, this phase appears with acicular shape with a color intermediate between the bright intense white points of ZrO2 and the blurred white color of the CaO. Point 3: ZrO2 nanoparticles.  Circular morphology, bright intense white color.

Figure 3. SEM images. (a-b) 100 wt. % MgO samples and (c-d) specimens MgO containing 5 wt. % of ZrO2 nanoparticles, all sintered at 1650°C. (e-f) 100 wt. % MgO and (g-h) MgO containing 5 wt. % of ZrO2 nanoparticles, all sintered at 1550 °C. (i) shows the shape of each of the phases identified as point 1 (CaO), point 2 (CaZrO3), point 3 (ZrO2). Figure 3 (a, c, e, and g) correspond to samples pressed by CUP. Figure 3 (b, d, f, and h) correspond to samples pressed by a CUP + CIP.

  1. In Fig.3g, no CaO/CaZrO3 phase was detected by SEM in XL4 ? Why?

A new micrograph was added for the XL4 sample where we clearly indicated the presence of the indicated phases CaO (point 1), CaZrO3 (point 2) and ZrO2 (point 3) and the matrix of MgO (see, Figure 3 (i)).

  1. The authors indicated that the pinning effect of ZrO2 and CaZrO3 particles reduced porosity and increased density of the MgO-ZrO2 ceramics. Please explain the reason of the pinning effect on porosity and density. Moreover, the grain size of MgO with ZrO2 is higher than that of MgO without ZrO2.

We appreciate your comment. A wide explanation is detailed as follow:

It is well known that the pinning effect contributes to the porosity elimination, and thus higher densification is obtained. The pinning effect (through ZrO2 and CaZrO3) avoids the grain boundary movement, allowing the porosity elimination. However, that phenomenon does not mean that the grain size cannot grow since all that is happening during sintering is cohesion between particles that leads to the grain growth. That is why MgO grains grow through the pinning effect. Below we demonstrated this effect graphically.

  1. In Section 3.3, the authors wrote “Fig.4 shows the mechanical behavior of ...”. “mechanical behavior” is not appropriate here.

Thank you for the comment, we have modified the sentence in line 298-300:

Figure 4 shows the variation of the cold crushing strength of the magnesia samples reinforced with ZrO2 nanoparticles (1, 3, 5 wt. %) and the reference sample of MgO (100 wt. % of MgO).

  1. In Fig.6(d) and Fig.7(d), the concentration of slag element was detected by spot analysis? If so, the elemental concentration may be seriously affected by spot size and location(at grain boundary or in grain).

Both in the Figure 6 (d) and Figure 7 (d) represent the concentration of elements (different from those of the matrix) of the slag (Mn, Ca, Si, Al, Ti) as a function of the distance in height of the specimen, from 0 mm to 6 mm. In this test, we have made different point analyses at the same height (through a cross cut) and the average of different point analyses at the same height is represented in the dotted red line in the Figure 6 (e) and Figure 7 (e).

As the concentration of the elements of the slag could be affected by the spot size, we paid significant attention to the fact that in all the elemental analyses, the spot size was of 50 µm. Regarding the location, we paid significant attention to the displacement of the analysis to continue with a straight line in both cases (see Figure 6 (e) and Figure 7 (e)). Regarding to this study, we concluded that the corrosion resistance is controlled by the CaZrO3, which difficulties the advance of the slag in the refractory. Similarly, ZrO2 nanoparticles act as a barrier both at the triple points and the grain boundaries, avoiding the advance of silicomanganese slag in the magnesia refractory.

  1. In Conclusion Section, the three mechanisms for enhanced strength should not be listed in conclusion. Firstly, t→m phase transition toughening mechanism was well known. More importantly, whether t→m phase transition occurred during strength testing has not been confirmed in this study.

Thank you for the comment, these considerations were removed from the “conclusion section” and were moved to other sections.

Reviewer 3 Report

The manuscript entitled „MgO-ZrO2 ceramic composites for silicomanganese production” concerns the influence of zirconia particles on the corrosion resistance of magnesia materials. The Authors studied the materials thoroughly. The methods of the sample preparation are well described. The sintered samples were characterized by means of XRD, SEM, XPS. Additionally, the basic physical and mechanical properties are given.

The materials were than tested with silicomanganese slag and the cross-section of the samples were than observed to investigate the slag penetration depth.

The manuscript is well organized with logical progression. The Authors characterized the materials and their application value insightfully.

In my opinion, there are only minor aspects that should be reconsidered/reviewed before publication:

  1. Table 1 – What was the method of porosity determination? if it is described elsewhere ([45]?) it should be mentioned in the text.
  2. Table 1 – two decimals would me more than enough for the measured density and porosity.
  3. page 3, line 118 – “the powder mixtures were formed” (not conformed).
  4. page 4, line 170 – instead of “draw”/”drawing” – scheme.
  5. Page 5 – please, add an information concerning the ZrO2 phase in the XRD description.
  6. page 8, lines 290-299 – Both MgO and CaO stabilize tetragonal/cubic phase of ZrO2. Some researches claim that the MgO is stabilizing ZrO2 already in temperature of 800°C after appropriately long annealing. Before concluding the strengthening mechanism of the MgO-ZrO2 composites, the ZrO2 phases should be identified.
  7. In my opinion, the influence of the decreased porosity on the corrosion resistance should be also mentioned in the conclusions.

Author Response

Dear Reviewer, thank you for your insightful comments about the manuscript. We answer your comments below the answer.

The manuscript entitled “MgO-ZrO2 ceramic composites for silicomanganese production” concerns the influence of zirconia particles on the corrosion resistance of magnesia materials. The Authors studied the materials thoroughly. The methods of the sample preparation are well described. The sintered samples were characterized by means of XRD, SEM, XPS. Additionally, the basic physical and mechanical properties are given.

The materials were than tested with silicomanganese slag and the cross-section of the samples were than observed to investigate the slag penetration depth.

The manuscript is well organized with logical progression. The Authors characterized the materials and their application value insightfully.

In my opinion, there are only minor aspects that should be reconsidered/reviewed before publication:

  1. Table 1 – What was the method of porosity determination? if it is described elsewhere ([45]?) it should be mentioned in the text.

We have added a sentence in line 132-133, where we explain how the porosity is determined:

“Bulk density and porosity tests were developed in accordance with the ASTM-C830-00 Standard and as described in the same Standard”.

Reference [45] was moved to the text, because there it is possible a deep description of the method indicated in the ASTM-C830-00 Standard.

For further information, the procedure used to determine the porosity is:

With each sample, three different weights were obtained: dry weight, soaked weight, and suspended weight. In accordance with ASTM-C830-00 standard, the following was performed:

  1. Drying of the specimens
  2. “Dry weight” is obtained
  3. Vacuum is applied for 45 min
  4. Water is added to the vacuum chamber, for 30 min
  5. The recipient is opened, for 30 min is resting
  6. They are removed from the chamber and they are dried with linen cloth
  7. “Saturated weight” is obtained
  8. Specimens were immersed in water to obtain the “Immersed weight”
  9. Finally, the following equations are used considering a temperature factor, to obtain the bulk density and apparent porosity (%).

        ................................. 1

            ................................ 2

Table 1. Temperature Factor

Temperature (°C)

Temperature factor (°C)

15

0.9991

16

0.9989

17

0.9988

18

0.9986

19

0.9984

20

0.9982

21

0.998

22

0.9978

23

0.9975

24

0.9973

25

0.997

26

0.9968

27

0.9965

28

0.9962

29

0.9959

30

0.9956

  1. Table 1 – two decimals would me more than enough for the measured density and porosity.

Agree, the change was made and is in line 137.

  1. page 3, line 118 – “the powder mixtures were formed” (not conformed).

Agree, the change was made and is in line 143.

  1. page 4, line 170 – instead of “draw”/”drawing” – scheme.

Agree, the change was made and is in line 199 and 206.

  1. Page 5 – please, add an information concerning the ZrO2 phase in the XRD description.

Figure 2 was modified to facilitate the identification of the ZrO2. We have also added information in line 233-235, about the ZrO2 phase detected in the sintered specimens (ICDD 00-050-1089), whose crystalline structure was tetragonal.

Figure 2. XRD pattern of the sample 100 wt. % MgO (brown and blue line, 2 and 4) and MgO sample containing 5 wt. % of ZrO2 nanoparticles (red and pink line, 1 and 3) sintered at 1550 °C and 1650 °C, pressed by: (a) Cold uniaxial pressing and (b) Cold isostatic pressing after cold uniaxial pressing (CUP + CIP).

  1. page 8, lines 290-299 – Both MgO and CaO stabilize tetragonal/cubic phase of ZrO2. Some researches claim that the MgO is stabilizing ZrO2 already in temperature of 800°C after appropriately long annealing. Before concluding the strengthening

We have included some additional information about this matter in lines 109-116 and 227-235:

ZrO2 nanoparticles used in this manuscript had monoclinic structure. Zirconium oxide (ZrO2) is a phase that characterizes by exhibiting three polymorphic transformations. It is monoclinic from room temperature up to 1170 ºC, tetragonal from 1171 ºC to 2370 ºC and cubic above 2370 ºC and until the melting point (2715 ºC).

It is well known that there are several phases that stabilize the ZrO2 as the MgO, CaO, Y2O3, which are adequate to stabilize a single crystalline structure of ZrO2 that is to say, that would remain with a certain crystalline structure form room temperature to the melting point (without polymorphic changes (monoclinic-tetragonal-cubic, or vice versa) and, therefore, volume changes that could involve cracks in the parts).

As well, we have observed in this research that the ZrO2 particles were completely surrounded by MgO particles (matrix constituent), in agreement with the MgO-ZrO2 binary diagram (the ZrO2 tetragonal phase admits 10% of MgO and the cubic phase admits up to 27% MgO in solid solution). This way, when the composites were sintered at 1600 ºC, there was a change from monoclinic to tetragonal, and this crystallographic change was maintained down to the room temperature as the MgO acted as stabilizer. This can be checked in the X-ray diffraction results in Figure 2 as with 5 wt. % of ZrO2, peaks corresponding to the tetragonal ZrO2 were observed (ICDD 50-1089).

  1. mechanism of the MgO-ZrO2 composites, the ZrO2 phases should be identified.

See the answer to the previous comment.

  1. In my opinion, the influence of the decreased porosity on the corrosion resistance should be also mentioned in the conclusions.

Added in line 565-566

Round 2

Reviewer 2 Report

The submitted manuscript has been revised by the authors, and is now acceptable for publication.